# Cyclic Moisture Sorption and its Effects on the Thermomechanical Properties of Epoxy and Epoxy/MWCNT Nanocomposite

**DOI:** 10.3390/polym11091383

**Published:** 2019-08-23

**Authors:** Tatjana Glaskova-Kuzmina, Andrey Aniskevich, Jevgenijs Sevcenko, Anna Borriello, Mauro Zarrelli

**Affiliations:** 1Institute for Mechanics of Materials, University of Latvia, Jelgavas 3, LV-1006 Riga, Latvia; 2Institute for Polymers, Composites and Biomaterials, National Research Council of Italy, 80055 Granatello, Portici, Italy

**Keywords:** cyclic moisture sorption, epoxy, multiwall carbon nanotubes, dynamic mechanical analysis, glass transition temperature

## Abstract

The aim of this work was to reveal the moisture absorption–desorption–resorption characteristics of epoxy and epoxy-based nanocomposites filled with different multiwall carbon nanotubes (MWCNTs) by investigating the reversibility of the moisture effect on their thermomechanical properties. Two types of MWCNTs with average diameters of 9.5 and 140 nm were used. For the neat epoxy and nanocomposite samples, the moisture absorption and resorption tests were performed in atmospheres with 47%, 73%, and 91% relative humidity at room temperature. Dynamic mechanical analysis was employed to evaluate the hygrothermal ageing effect for unconditioned and environmentally “aged” samples. It was found that moisture sorption was not fully reversible, and the extent of the irreversibility on thermomechanical properties was different for the epoxy and the nanocomposite. The addition of both types of MWCNTs to the epoxy resin reduced sorption characteristics for all sorption tests, improved the hygrothermal and reduced the swelling rate after the moisture absorption–desorption.

## 1. Introduction

The addition of nanoparticles to traditional polymer resins allows the final material properties to be tuned by changing the concentration and morphology of the nanoparticles. Multiwall carbon nanotubes (MWCNTs) have unique mechanical properties; their stiffness and strength values are within the range of 100–1000 and 2.5–3.5 GPa, respectively, and their electrical conductivity is 3000–4000 S/m. Such properties make them valuable candidates to develop novel composites characterized as advanced polymer materials [1,2].

However, to date, such materials have been limited mostly to indoor applications due to the relative sensitivity of the mechanical properties of polymers and polymer composites to environmental conditions, such as moisture and temperature [3]. The analysis of the available literature revealed contradictory information on the reversibility of hygrothermal effects on the structure and mechanical properties of epoxy [3,4,5] and epoxy-based nanocomposite (NC) [6,7,8].

The reduction of the diffusion coefficient of water in epoxy filled with MWCNTs (0.3–1 wt %) was found as the temperature increased from 20 to 70 °C, while the levels of moisture/water uptake remained unchanged [7]. This was explained by lower free volume, while reduced rubbery modulus and tanδ peak were explained by higher crosslink density and lower segmental mobility in the NC [7].

Although, the equilibrium water content significantly decreased with the addition of carbon nanofibers (CNFs, 0.25–1 wt %) and MWCNTs (0.1–0.25 wt %) to the epoxy resin, the plastization effect was greater, leading to a similar glass transition temperature for the neat epoxy and epoxy/CNF, and an almost two-fold decrease for epoxy/MWCNT [6]. This effect was explained by the location of MWCNTs within the free volume of the epoxy network as well as by the water-induced plastization and post-curing, acting synergistically.

Almost the same effect related to moisture uptake was observed on the tensile elastic properties of the epoxy and epoxy/MWCNTs (0.3–1 wt %) composites for samples conditioned up to saturation [8]. These effects induce a decrease of elastic modulus (~5%–8%) and strength (~18%–22%) compared to the initial state. Moreover, when the temperature was increased from 20 to 70 °C, a non-monotonic decrease in the strength was revealed, likely due to the catalytic effect of the water during the post-curing reaction which acts as plasticizer at elevated temperatures [8].

The maximal improvements in barrier characteristics and stability against environmental factors could be obtained by optimizing the dispersion process and the chemical functionalization of MWCNTs in order to reduce the nanoparticles agglomeration, improving at the same time the interfacial strength with surrounding polymer matrix [9]. Moreover, nanotube degree of dispersion depends also on the uniformity of the boundary layer and considered nanotube morphology [10]. For all these reasons, the number of factors affecting the final performance of the NC can be rather considerable leading to contradictory results towards the improvements in barrier and mechanical properties discussed before.

A better understanding of moisture diffusion phenomena in polymers filled with different nanoparticles and the development of structure-properties relationships for them has fundamental importance for their practical applications. Thus, the consideration of cyclic environmental effects and the identification of the environmental stability of NC may broaden their application to outdoor conditions with a wide range of temperature and moisture values.

The novelty of the research was to find the interrelation among the absorption–desorption–resorption characteristics in a wide range of humidity and the thermomechanical properties before and after environmental ageing for the epoxy filled with different MWCNTs.

The aim of this work was to estimate the peculiarities of the moisture absorption–desorption–resorption of epoxy and epoxy-based NC filled with different MWCNTs and to determine the reversibility of the moisture effect on their thermomechanical properties. For this purpose, moisture ab/de/re-sorption tests were performed in atmospheres with different relative humidity, and dynamic mechanical analysis tests were carried out in an unconditioned state and after the moisture absorption–desorption cycle for the epoxy and the NC.

## 2. Materials and Methods

A commercially available monocomponent RTM6 (Hexcel Composites, Stamford, CT, USA) epoxy resin was used as a matrix material. This system is characterized by a low viscosity and a glass transition temperature *T*_g_ of about 200 °C. It is used for infusion processes, such as resin transfer molding or vacuum bag infusion, for both industrial and academic purposes.

Two different MWCNT grades were used as nanofillers: (1) SA659258 (Sigma-Aldrich, St. Louis, MO, USA) with diameters of 110–170 nm, lengths of 5–9 µm, and a carbon purity of ≥90% [11] and (2) N7000 (Nanocyl, Sambreville, Belgium) with an average diameter of 9.5 nm and length of 1.5 µm and a carbon purity of 90% [12]. The nanofillers were denoted as “SA” (Sigma Aldrich) and “N” (Nanocyl). Thus, the average aspect ratios of SA and N evaluated by using data from the Technical Datasheets of the suppliers were about 50 and 150, respectively. Different contents of the SA and N nanofillers were used, ranging from 0.005 to 2 wt %. Considering previous experience with the dispersion of the same nanofillers, different loadings were added to the hosting matrix for N (0.005–0.01 wt %) and SA (0.1–2 wt %) [9,13], achieving the maximal possible nanofiller content and satisfactory dispersion at the same time. The choice of the nanofillers was based on similar configuration (multiwall CNTs) and carbon purity but different aspect ratio and filler content. Though, the nature of these nanofillers, as well as their morphology and loading are meaningful parameters that could influence the performance of the NC, the use of such different nanofillers may allow investigating kinetics of cyclic moisture sorption in a wider range of filler loadings.

The manufacturing procedure of the NC consisted of the following steps: (1) ultrasonication of the nanofiller and epoxy resin using a dipping tip sonicator (Misonix S3000, Farmingdale, NY, USA) for 60 min at a power output of 20 W and a temperature of 120 °C; (2) degassing for 30 min at 80 °C; (3) curing for 1 h at 160 °C and post-curing for 2 h at 180 °C. The processing method was properly designed to produce a good dispersion of pristine nanotube aggregates [13]. For each filler content, two discs, each with a diameter of 100 mm and a thickness approximately of 2.5 mm, were obtained and cut into bar-shape specimens with average dimensions of 70 ± 2, 10 ± 1, and 2 ± 0.2 mm. Prior to moisture sorption tests, all samples were polished to get equal dimensions and to reduce the effect of surface roughness on the moisture sorption kinetics.

The dispersion degree of the MWCNT in the epoxy resin on the microscale was investigated with an optical microscope (Olympus BX51, Tokyo, Japan). Three images were obtained for each material type, and these were used for the analysis. According to Figure 1, the MWCNTs of both aspect ratios (N and SA) were dispersed rather homogeneously and with a minimal amount of large agglomerates on the microscale, which may have negatively affected the sorption and thermomechanical properties investigated in this study.

Before starting the moisture absorption tests, all specimens were conditioned in a desiccator with silica gel at a relative humidity of 24% to remove the absorbed moisture until all specimens showed no change in mass (hereinafter, the initial state). This was reached by all samples in approximately 2 months at room temperature, and the amount of moisture desorbed by all samples was 1.0% ± 0.1%. Thus, equal conditions were ensured for the epoxy and NC samples before starting the moisture absorption tests in atmospheres with different humidity levels. Subsequently, the test samples were placed in desiccators at relative humidity of 47%, 73%, and 91% (absorption), which were achieved by using saturated solutions of KCNS, NaCl, and K_2_SO_4_, respectively. Such conditioning allowed investigating the sorption kinetics of the epoxy and the NC in a wide range of humidity. After all samples had reached the equilibrium moisture content, they were dried in an atmosphere with 24% relative humidity (RH) until no change in mass was reached (desorption). Then, they were moistened in the appropriate humid atmosphere again (resorption). Thus, for all samples, the cycles of moisture absorption and desorption in the atmosphere with relative humidity of 47%, 73%, and 91% were indicated with following labels: 24-47-24, 24-73-24, and 24-91-24, accordingly. Moisture ab/de/re-sorption experiments were performed at room temperature (*T* = 20 °C). For the investigation of the moisture sorption kinetics, the specimens were periodically removed from the desiccator and weighed by using a Mettler Toledo (Columbus, OH, USA) XS205DU balance with a precision of 0.05 mg. Swelling measurements during absorption and resorption were performed by using a Mitutoyo (Sakado, Japan) gauge strand 215-151-10. At least five specimens were used for each material type, and the averaged values of the replicates together with their standard deviations (as error bars) are presented.

DMA tests were performed in tensile configuration at a force of 4 N by using a Mettler Toledo (Columbus, OH, USA) DMA/SDTA861 instrument to evaluate the hygrothermal ageing effects in both initial and “aged” NC samples after the moisture absorption–desorption cycle. All experiments were performed at a frequency of 10 Hz from 30 to 280 °C with a 3 K/min heating rate to assure slow heating, and thus avoiding any temperature gradient. The maximum peak of the tanδ vs. *T* plots was used to identify the α-relaxation associated with the glass transition and the corresponding temperature was assumed to evaluate *T*_g_ for all tested samples. For each initial and environmentally “aged” sample, the nominal dimensions were 30 × 3 × 1 mm^3^, and no less than two scans were performed. The glass transition temperature of the samples was evaluated as the averaged value of two scans with the error bars representing their standard deviations.

## 3. Results and Discussion

### 3.1. Cyclic Moisture Sorption

For the description of moisture absorption curves of epoxies and epoxy-based composites, the classical Fick’s model is usually applied [4,5,7,14,15,16]. In the case of 3D mode, the moisture content *w*(*t*) in the specimen is expressed as
(1)$$w(t)=w_{\infty }-\left(w_{\infty }-w_{0}\right)\frac{8}{\pi^{6}}\sum_{k=1}^{\infty }\sum_{n=1}^{\infty }\sum_{m=1}^{\infty }\frac{{\left[1-{\left(-1\right)}^{k}\right]}^{2}{\left[1-{\left(-1\right)}^{n}\right]}^{2}{\left[1-{\left(-1\right)}^{m}\right]}^{2}}{k^{2}n^{2}m^{2}}\exp\left[-\lambda^{2}{}_{k,n,m}Dt\right]$$
where $\lambda_{k,n,m}^{2}=\lambda_{k}^{2}+\lambda_{n}^{2}+\lambda_{m}^{2}={\left(\frac{\pi k}{a}\right)}^{2}+{\left(\frac{\pi n}{b}\right)}^{2}+{\left(\frac{\pi m}{l}\right)}^{2}$; *w*_0_ and *w*_∞_ are the initial and equilibrium moisture contents; and *a*, *b,* and *l* are respectively the thickness, width, and length of the specimen. *D* identifies the diffusion coefficient of the material.

Basically, in Equation (1), there are two independent parameters: the diffusion coefficient *D* and the equilibrium moisture content *w*_∞_. The equilibrium moisture content is the maximal achieved moisture content over the sorption test, while the time-varying moisture content (in %) is obtained by using the following expression:(2)$$w(t)=\frac{m(t)-m_{0}}{m_{0}}\cdot 100$$
where *m*(*t*) is the time-varying mass of the specimen at time *t*, and *m*_0_ is the mass of the specimen in the initial state (*t* = 0). Then, the diffusion coefficient is calculated from the initial slope of the curve *w* vs. $\sqrt{t}$ [17]:(3)$$D=\frac{\pi h^{2}}{16t}{\left(\frac{w(t)-w_{0}}{w_{\infty }-w_{0}}\right)}^{2}$$

Usually, Fick’s model provides a satisfactory description of the experimental data for the moisture absorption of epoxies and epoxy-based composites at room temperature [4,7,14,15,16]. However, as a matter of fact, at elevated temperature and humidity, non-Fickian diffusion starts to dominate since the polymer network may be significantly “disturbed”, and large adjustments to segmental conformations are required to reach equilibrium with long relaxation times [4].

One of non-Fickian models which may improve the description of the experimental data at high relative humidity is the model with a time-variable diffusivity [15,16,18]. According to this model, due to different physical processes (e.g., the plasticization, ageing, post-cure etc.), the diffusion coefficient decreases with time:(4)$$D(t)=D_{0}\cdot \exp(-\gamma t)$$
where *D*_0_ is the diffusion coefficient at the initial instant of time, but *γ* is the coefficient describing the rate of change for it. Then, according to this model the expression for determining the moisture content in the specimen for the three-dimensional case takes the form [16,18]
(5)$$\begin{array}{l} w(t)=w_{\infty }-\left(w_{\infty }-w_{0}\right)\frac{8}{\pi^{6}}\sum_{k=1}^{\infty }\sum_{n=1}^{\infty }\sum_{m=1}^{\infty }\frac{{\left[1-{\left(-1\right)}^{k}\right]}^{2}{\left[1-{\left(-1\right)}^{n}\right]}^{2}{\left[1-{\left(-1\right)}^{m}\right]}^{2}}{k^{2}n^{2}m^{2}}\\ \times \exp\left[-\lambda^{2}{}_{k,n,m}\times \frac{D_{0}}{\gamma }\left[1-\exp\left(-\gamma \cdot t\right)\right]\right] \end{array}$$
where *γ* = 1/*τ*, and *τ* is the characteristic time of relaxation. Thus, the model contains three parameters—the diffusion coefficient at the initial instant of time *D*_0_, the equilibrium moisture content *w*_∞_, and the coefficient *γ* describing the rate of change in the diffusion coefficient.

According to [6,7,9,19] 1D Fick’s model provided a satisfactory description of experimental data for moisture absorption by the epoxy filled with MWCNTs. In the current study the purpose was to evaluate if the Fick’s model correlates well with the experimental data also for the case of cyclic moisture absorption: moisture absorption, desorption, and resorption. It is worth noting that preliminary evaluation by using 1D, 2D, and 3D models (Equations (1) and (5)) has shown that the moisture sorption curves evaluated by three-dimensional models were the closest to the experimental data. Therefore, hereinafter they were used in the description of moisture kinetics for the epoxy filled with N and SA during moisture absorption, desorption, and resorption.

The moisture absorption, desorption, and resorption results for the epoxy resin in an atmosphere with 91% RH are presented in Figure 2a. To improve the description of experimental data for moisture absorption both Fick’s model and the model with time-variable diffusion coefficient were applied. Obviously, the application of the model with time-variable diffusion coefficient provided better description of the experimental data than the Fick′s model and were used for further analysis of the sorption characteristics during the absorption. It is seen that all sorption processes were characterized by similar trends with a considerably higher sorption rate in the early stages and a progressively slower water uptake as the equilibrium plateau was approached. Similar results were obtained for the epoxy filled with both N and SA. Figure 2b and c reveal the effects of N and SA nanotubes on moisture absorption and resorption kinetics. Obviously, as shown in Figure 2a the sorption process in the epoxy (as well as in the NC) was not fully reversible. The rate of moisture diffusion was the highest for desorption, the lowest for resorption, and at intermediate level for the absorption process. Similar results were discussed in [4] for desorption and resorption, which were shown to occur faster than absorption. Moreover, the difference among predictions computed by Fick′s model and the experimental data vanished comparing moisture absorption and resorption.

The fact that resorption occurred faster than absorption and desorption could be explained by the increase in the free volume of a polymer matrix that occurs due to the swelling effect [4,20]. Basically, the free volume in a polymer may be considered as a difference between the measured and occupied volumes of the polymer and its content in a polymer may change over a long period of time due to physical ageing, thus affecting the overall properties of the polymer. Nevertheless, almost the same or a slight increase in the equilibrium moisture content was observed for all sorption processes, due to the hygrothermal ageing of the epoxy [4]. According to Figure 2a, desorption was characterized by the highest rate of diffusion and the highest equilibrium moisture content in comparison with absorption and resorption, both for the epoxy and the NC with SA and N, which is in line with the swelling effect discussed previously. However, the moisture absorption–desorption cycle might induce irreversible structural changes and subsequently might change the amount of free volume available for water molecules to diffuse through the material. Therefore, the resorption process may occur differently than absorption with a different diffusion coefficient and equilibrium moisture content.

The effect of N and SA loading on the kinetics of moisture absorption and resorption of the epoxy is shown in Figure 2b,c, respectively. Obviously, a slight reduction of the equilibrium moisture content both for absorption and resorption processes was obtained for the NCs. The sorption isotherms given in Figure 3 reveal a slight reduction of equilibrium moisture content for the case of the NC compared to the epoxy. The maximal reduction of the equilibrium moisture content in comparison with the epoxy resin was observed in atmosphere with 91% RH: for the epoxy filled with 0.1 wt % of N it was −0.11%, while for the epoxy filled with 2 wt % of SA it was −0.21%, respectively.

The diffusion coefficient which was calculated by Equation (3) in all atmospheres was 7.3 ± 0.2 × 10^3^ mm^2^/h for the epoxy and 6.6 ± 0.2 × 10^3^ mm^2^/h for the epoxy filled with N and SA MWCNTs. This value was used as *D*_0_, the diffusion coefficient at the initial instant of time, for the model with time-variable diffusion coefficient. The coefficient of proportionality *γ* for the epoxy and all NCs during moisture absorption was almost the same: 0.0004 ± 0.0001 and 0.0003 ± 0.0001, accordingly. In general, due to the presence of three parameters, the model with a time-variable diffusivity was relatively flexible and provided better description of the sorption curves in comparison with Fick′s model. For sure, such a small reduction in the sorption characteristics for the NC can be explained by the high quality, mechanical properties, and resistance to moisture absorption of the epoxy, which is widely applied for aeronautical elements in liquid infusion processes. Similar results and observations were obtained and discussed for Araldite LY556 (Sigma-Aldrich, St. Louis, MO, USA) filled with different types of graphene nanoplatelets [21].

Generally, the reduction in sorption characteristics of the polymers could be explained by assuming a reduction in the free volume within the polymer due to the addition of different types of nanoparticles, particularly exfoliated platelet-shape nanoparticles [7,22]. This might occur mostly due to enhanced physical ageing, differences in cure kinetics, and the segmental mobility of polymer chains upon the addition of the nanofillers.

It is possible to perform indirect evaluation of the changes in the free volume in polymers before and after the absorption–desorption cycle, by calculating the swelling strain, computed as relative change in the length of the test specimens, as follows:(6)$$\varepsilon_{sw}=\frac{l(t)-l_{0}}{l_{0}}\cdot 100$$
where *l*(*t*) is the time-varying length of the wet specimen at time *t*, and *l*_0_ is the length of the specimen in the initial (dry) state (*t* = 0).

The swelling strain as a function of the moisture content in atmospheres with different relative humidity (47%, 73%, and 91%) is shown in Figure 4 for the epoxy and the epoxy filled with 0.1 wt % of N and 2 wt % of SA. Obviously, by increasing the relative humidity of the atmosphere, the swelling strain increased as well. However, the relationship between the swelling strain and equilibrium moisture content for the absorption and resorption processes was slightly different for the epoxy and the NC.

It can be hypothesized that a higher reduction in the rate (slope) of swelling of the NC in comparison with the epoxy could be related to the increased contribution of “bound” water and a subsequent decrease in the free volume after a cycle of moisture absorption–desorption. Similar ideas were formulated, and results obtained in [7] where the neat epoxy and its NC with the content of 0.3–1 wt % of MWCNTs were characterized by nearly the same swelling coefficient $\beta =\Delta \varepsilon_{sw}/\Delta w_{\infty }$ which was equal to 0.24. Herein, the estimation of the swelling coefficient as shown in Figure 3, gave different results and allowed the effect of the nanofiller content to be revealed. The epoxy resin had a β value of 0.24 ± 0.02 for absorption and 0.23 ± 0.02 for resorption. While the addition of 0.1 wt % of N resulted in the reduction of the swelling coefficient β from 0.28 ± 0.02 to 0.25 ± 0.05, and the addition of 2.0 wt % of SA resulted in a reduction in β from 0.25 ± 0.02 to 0.19 ± 0.02. Thus, the epoxy filled with 2.0 wt % of SA showed the greatest decrease of the swelling coefficient if compared with the epoxy and the epoxy filled with 0.1 wt % of N

Moreover, the diffusion coefficient and equilibrium moisture content in a polymer NC is a function, to a large extent, of the interplay between the free volume content and the tortuosity factor of the nanoparticles [22]. The tortuosity of the nanoparticles is dependent on many factors, such as the morphology of the dispersed nanoparticles, their arrangement and orientation in the diffusion direction, and their volume fraction [23]. The result of this interplay (increase or decrease of D) upon the addition of nanoparticles determines which of the two factors would become predominant.

To better understand the differences in diffusion phenomena in the epoxy, filled with different aspect ratio MWCNTs, Nielsen model could be applied considering parallel periodic arrangement of straight filler particles. According to this model the tortuosity (*τ*) of any filler particles could be calculated by the following equation [22,24]:(7)$$\tau =1+\frac{\alpha }{2}\cdot \nu$$
where α=l/h is the aspect ratio (length *l* vs. thickness *h*) of the nanoparticles, and *v* is the filler volume content which can be estimated by using equation [25]:(8)$$v=\frac{\rho_{m}\cdot c}{\rho_{m}\cdot c+\rho_{f}\cdot \left(1-c\right)}$$
where *c* is the filler weight fraction, and $\rho_{f}$ and $\rho_{m}$ are the density of the filler and the polymer matrix, respectively. To compare current results of theoretical tortuosity of MWCNTs in the epoxy resin with the literature data, two additional nanocomposites were chosen: epoxy resin, respectively, filled with NC3152 (Nanocyl, Sambreville, Belgium) having aspect ratio of 100 [6] and C150P (Bayer, Leverkusen, Germany) having aspect ratio of 77 [7]. The density of the epoxies and the MWCNTs were taken from the material datasheets [11,12,26,27,28,29], and instead the filler content was kept the same as in [6,7].

The results of the calculation of the tortuosity factor according to Equation (5) for the epoxies filled with MWCNTs having different aspect ratios are reported in Figure 5. Obviously, N and NC3152, which have higher aspect ratio, are characterized by higher coefficient of proportionality compared to SA and C150P. Indeed, if it would be possible to increase the NA and NC3152 filler content, regardless of the viscosity increase for the final NC, the tortuosity factor would be much higher at the same filler content. However, due to high viscosity of the NC filled with the MWCNTs of high aspect ratio at a filler content of more than 0.1–0.25 wt %, this can be hardly achieved. As shown in Figure 4 for the epoxy resin filled with N (α ≈ 158) at filler content ranging from 0.003 to 0.065 vol % (i.e., 0.005–0.1 wt %) the tortuosity factor resulted within the range 1.0–1.05, whereas for the same epoxy resin filled with SA (α = 50) at a filler content within the range 0.34–1.35 vol % (i.e., 0.5–2 wt %), it would be much higher—1.08–1.34.

Finally, the comparison of N and SA nanotube types shows that for SA filler with an aspect ratio three times smaller than N-type nanotube, at maximum content percentage (i.e., 2 wt %) is characterized by an approximately 27% higher tortuosity factor due to the increase in filler content by 21 times. Evidently, the effects on the sorption characteristics such as diffusion coefficient and equilibrium moisture content are the highest for the epoxy resin filled with 2.0 wt % of SA (maximum filler content).

### 3.2. Thermomechanical Properties

Dynamic mechanical analysis (DMA) is widely used to explore the viscoelastic properties of polymers that occur as a response to the application of oscillating forces and make conclusions about mobility of polymer chains and the deterioration of polymers’ thermomechanical properties due to hygrothermal ageing effects [6,7].

To evaluate the reversibility of the moisture absorption effect on the dynamic mechanical properties (storage modulus *E*’ and loss factor tan*δ*) of the epoxy and epoxy filled with N and SA, these characteristics were analyzed in the initial state and after the moisture absorption–desorption cycle in atmospheres with relative humidity levels of 47%, 73%, and 91%. The representative DMA curves for unaged and environmentally aged specimens after a cycle of moisture absorption–desorption in atmospheres with different relative humidity values are shown in Figure 6, respectively, for the epoxy (a) and the epoxy filled with 0.1 wt % of N (b) and 2 wt % of SA (c), respectively.

It is worth to notice from Figure 6, that the cycle of moisture absorption–desorption in different atmospheres caused irreversible changes to the thermomechanical properties of all materials, and these changes varied for the epoxy and the NCs. As shown in Figure 7, for the epoxy, the glass transition temperature increased from 226 °C (in the initial state) to 234 °C (after moisture cycle 24-91-24); for the epoxy filled with 0.1 wt % of N, it decreased from 236 °C (in the initial state) to 230 °C (after moisture cycle 24-91-24); and for the epoxy filled with 2 wt % of SA, it also decreased from 235 °C (in the initial state) to 233 °C (after moisture cycle 24-91-24).

Rather contrary results for DMA curves and *T*_g_ were published in the literature. Almost total reversibility of the plastization was obtained by water after desorption, causing to return to the same glass transition temperature as that of epoxy polymer under the initial conditions. This was explained by the reformation of secondary bounds [30,31]. Though, in the case of non-complete curing of polymers, especially at high temperatures, post-curing phenomena can occur during water absorption, leading to no change or even an increase in *T*_g_, proving that the post-curing process dominates over plastization [5,6], especially for cold-cured polymers [20,32].

Herein, according to Figure 7 the glass transition temperature of the epoxy increased with the higher desorbed moisture content after the absorption–desorption cycle likely due to possible post-cure and decreasing of the free volume, as previously discussed. Similar results of *T*_g_ increasing after moisture absorption were attributed to the post-curing phenomenon during physical ageing for a partially cured system [5]. The minimal effect of the addition of both N and SA on the glass transition temperature is fully in line with available literature and it can be attributed to several contrasting factors such as the existence of interphases/interfaces on the surface of the nanoparticles, agglomerates, changes in crystallinity, and the crosslinking density of the epoxy resin [7,33].

Moreover, it can be noticed that for the epoxy resin, the height of tanδ curves decreased after the moisture absorption–desorption cycle, which can be attributed to lower degree of mobility of polymer chains caused by microstructural reorganization during hygrothermal ageing and the subsequent decrease of the free volume in the epoxy resin. This result correlates well with the reduced diffusion coefficient and equilibrium moisture for moisture resorption in comparison with moisture absorption for all materials studied. For the NC, this effect was not so pronounced, apparently due to the possible reduction of free volume (in comparison with the epoxy resin) in the initial state caused by enhanced physical ageing. Similar observation and ideas regarding a difference in cure kinetics of the polymers and NC, as well as the reduced segmental mobility of the polymer chains were formulated in [7,22,23,34].

The effect of moisture absorption–desorption on the storage modulus can be observed as an overall slight decrease in both glassy and rubbery regions for all materials. It can be seen from Figure 6 that the higher the relative humidity for the sorption process was, the lower is the storage modulus in the desorbed state was for all materials. The maximal decrease in the storage modulus was observed after desorption in the atmosphere with a relative humidity of 91%. For the epoxy resin, at 30 and 280 °C, the storage modulus decreased by 15% and 13%, but for the epoxy resin filled with 0.1 wt % N, it decreased by 3% and 9%, and for the epoxy filled with 2 wt % SA, it decreased by 9% and 10%, respectively. These results correlate well with the reduced sorption characteristics and considerations about the reduced free volume for the NC.

## 4. Conclusions

In this study, the characteristics of moisture absorption–desorption–resorption of epoxy and epoxy-based NC filled with different MWCNTs were estimated and the reversibility of the moisture effect on their thermomechanical properties was determined. It was experimentally verified that moisture sorption was not fully reversible, and the extent of the irreversibility on thermophysical properties was variable comparing the epoxy and the corresponding NC at different humidity levels.

In comparison with moisture absorption and resorption, desorption had the highest rate of diffusion and the highest equilibrium moisture content in both epoxy and NC, for all considered aspect ratios MWCNTs. This conclusion was supported also by the reduction in the swelling rate and explained by the possible irreversible structural changes (viz. reduction of the free volume) during the cycle of moisture absorption–desorption.

In all sorption tests, the epoxy, filled with both N and SA, had a lower diffusion coefficient and equilibrium moisture content than the neat epoxy, obviously due to the reduced mobility of polymer chains and the lower free volume available allowing the water molecules to diffuse through the material. The theoretically estimated tortuosity factor of SA MWCNTs was much higher (by 27%) than for N MWCNTs due to their much higher filler content, leading to a greater effect on the reduction of all sorption characteristics.

The moisture absorption–desorption cycle resulted in irreversible changes also in the thermomechanical properties. Contrary results were obtained for the epoxy and the NC. The glass transition temperature slightly increased for the environmentally “aged” epoxy resin and decreased for the epoxy filled with N and SA. For all materials, the greatest effect was observed after moisture absorption–desorption cycle in the atmosphere with the highest relative humidity (91%). The increase in the glass transition temperature of the epoxy can be hypothetically explained by the lower degree of mobility of the polymer chains caused by microstructural reorganization during hygrothermal ageing and the subsequent decrease of the free volume in the epoxy resin. Minor variations of the glass transition temperature of the NC in comparison with the neat polymer resin were found. The effect of moisture absorption–desorption on the storage modulus can be observed as an overall slight decrease in both glassy and rubbery regions for all materials studied with a slightly higher decrease for the epoxy resin. These results correlate well with the reduced sorption characteristics and the free volume considerations for the epoxy filled with N and SA, resulting in slightly improved hygrothermal stability in comparison with the neat epoxy resin.

## Figures and Tables

**Figure 1 polymers-11-01383-f001:**
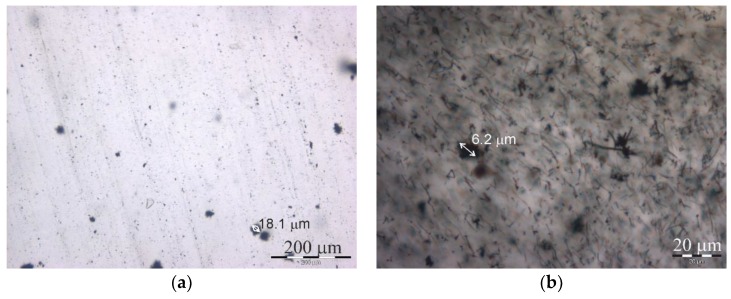
Optical micrographs of the epoxy filled with multiwall carbon nanotubes (MWCNT) contents of 0.005 wt % for N (**a**) and 1 wt % for SA (**b**).

**Figure 2 polymers-11-01383-f002:**
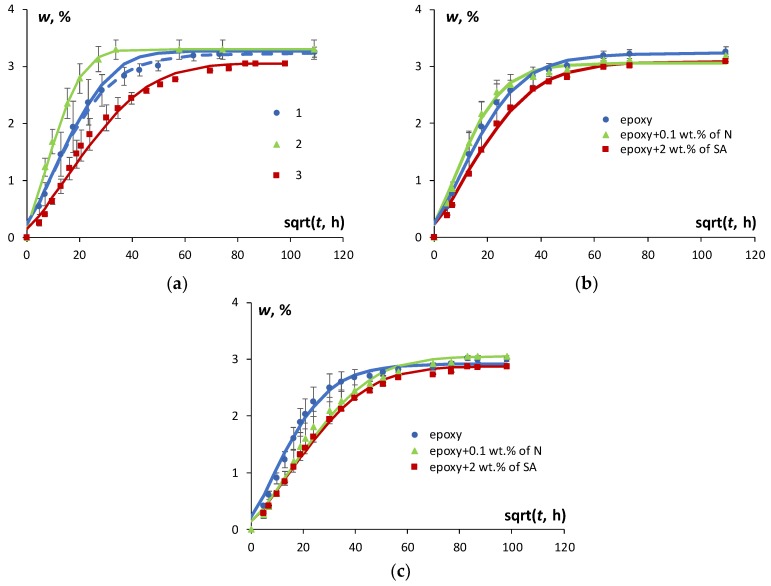
Moisture kinetics of epoxy specimens for moisture absorption (1), desorption (2), and resorption (3) at a relative humidity of 91% during moisture absorption and resorption and 24% during desorption (**a**). Moisture absorption (**b**) and resorption (**c**) for the epoxy and the epoxy filled with 0.1 wt % of N and 2 wt % of SA in an atmosphere with 91% relative humidity (RH). Dots—experimental data, lines—calculation by Equations (1) (a–solid, c) and (5) (a–dashed, b).

**Figure 3 polymers-11-01383-f003:**
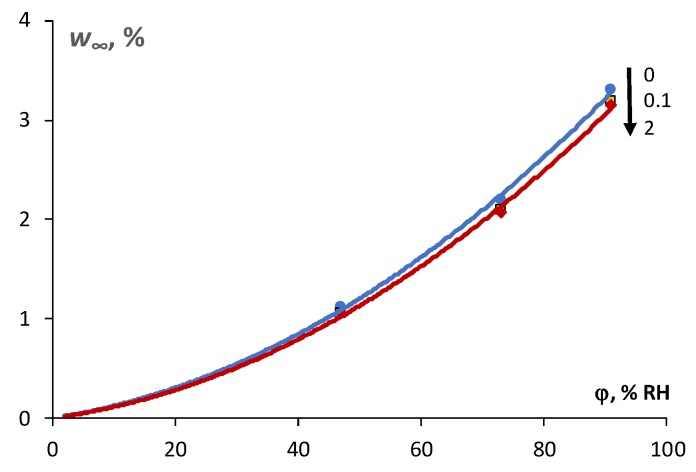
Sorption isotherms of the epoxy and the nanocomposite (NC) during moisture absorption in atmosphere with 91% RH (filler content is indicated in the legend).

**Figure 4 polymers-11-01383-f004:**
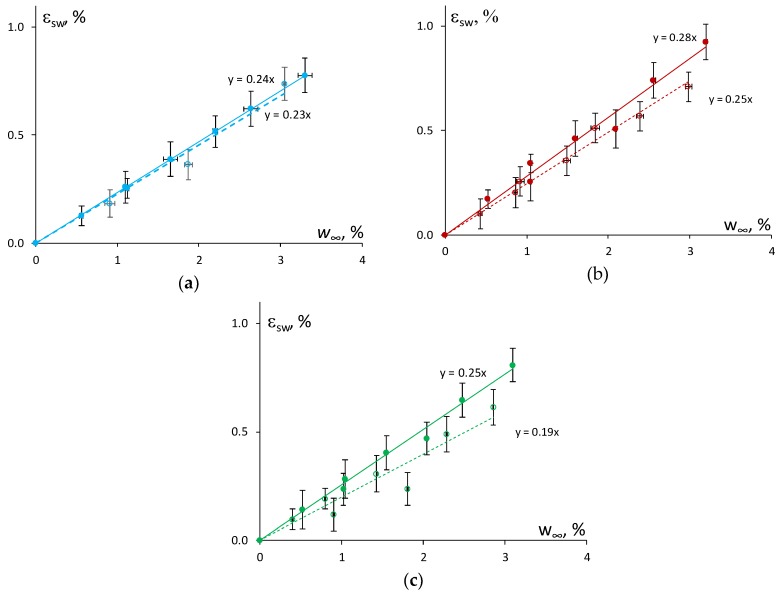
Swelling strain vs. moisture content for the absorption (solid line) and resorption (dashed line) of the epoxy (**a**) and the epoxy filled with 0.1 wt % of N (**b**) and 2 wt % of SA (**c**).

**Figure 5 polymers-11-01383-f005:**
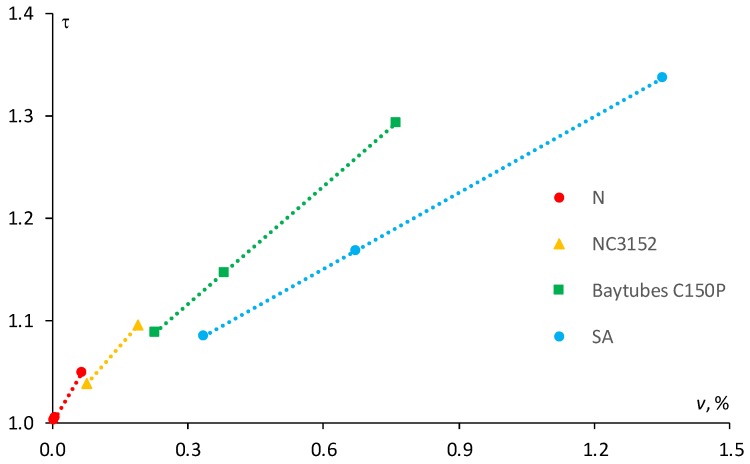
Tortuosity factor vs. the filler volume fraction for the epoxy filled with different MWCNTs (indicated in the legend). Dots—calculation by Equation (5), lines—linear approximation.

**Figure 6 polymers-11-01383-f006:**
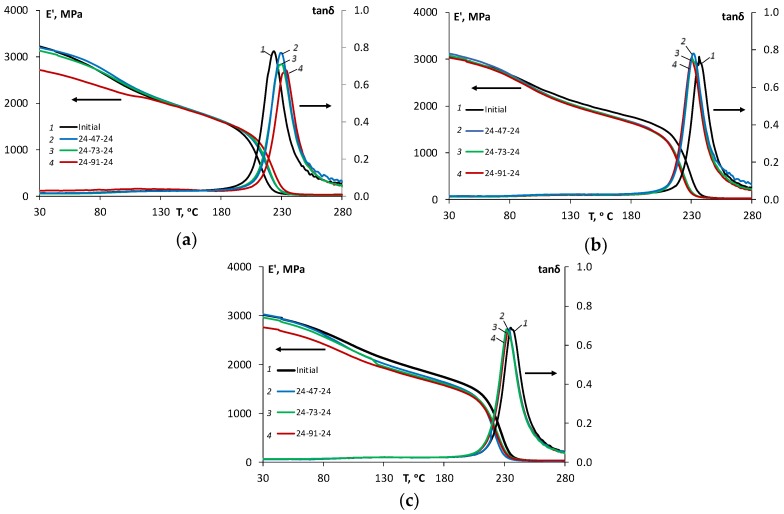
Storage moduli (*E*’) and loss factor (tanδ) vs. temperature for the epoxy resin (**a**), epoxy filled with 0.1 wt % of N (**b**) and 2 wt % of SA (**c**), in the initial state and after moisture absorption–desorption in atmospheres with 47%, 73%, and 91% RH (indicated in the legend).

**Figure 7 polymers-11-01383-f007:**
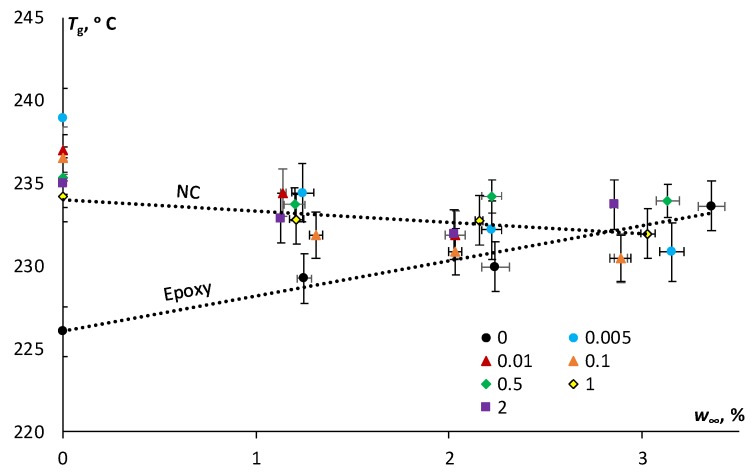
Glass transition temperature vs. desorbed moisture content after moisture absorption for epoxy and epoxy filled with N and SA MWCNTs (filler content is indicated in the legend). Dots—experimental data, lines—linear approximations for the epoxy and the NCs.

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
