# Peer review of "Cyclic Moisture Sorption and its Effects on the Thermomechanical Properties of Epoxy and Epoxy/MWCNT Nanocomposite"

_polymers, 2019, doi:10.3390/polym11091383_

Round 1

Reviewer 1 Report

This paper aims to investigate the moisture absorption–desorption–resorption characteristics of epoxy and epoxy-based nanocomposites filled with multiwall carbon nanotubes (MWCNTs) with different aspect ratios by investigating the reversibility of the moisture effect on their thermomechanical properties. Two different typologies of nanotubes were considered at three humidity conditions. Full characterization and interesting results were reported.

The publication has technical merit and it is worth publishing, however, some spelling mistakes and readjustments are needed to comply with the journal standards.

Some of the links within the manuscript should be changed as error referred.

Please revised Figure captions with missing links

I would suggest minor revision for this manuscript indicating that the discussion and the reference reported for the assumption during the data analysis are very valuable and congruent with the overall phenomenon description involving structural rearrangement and physical ageing of the analysed epoxy and nanoepoxy samples.

Author Response

1. The publication has technical merit and it is worth publishing, however, some spelling mistakes and readjustments are needed to comply with the journal standards.

English language was edited. The corrections are indicated in the text by using track changes.

2. Some of the links within the manuscript should be changed as error referred.

The links were checked and corrected.

3. Please revise Figure captions with missing links

The link of the caption (Fig. 4) was corrected.

Reviewer 2 Report

The presented article is interesting and a construction of article is logical. Scientific merit is good. The work is relevant and practical. Clarity of expression and communication of ideas, readability and discussion of concepts is medium. Sufficient discussion of the context for the work and suitable referencing is very  good. Article:  Cyclic Moisture Sorption and its Effects on the Thermomechanical Properties of Epoxy and Epoxy/MWCNT Nanocomposite is very interesting and has practical use. Methodology is well chosen. The authors presented in this paper The aim of this work was to reveal the moisture absorption–desorption–resorption  characteristics of epoxy and epoxy-based nanocomposites filled with multiwall carbon nanotubes  (MWCNTs) with different aspect ratios by investigating the reversibility of the moisture effect on  their thermomechanical properties. Two types of MWCNTs with average diameters of, respectively, 140 and 9.5 nm were used. For the epoxy and nanocomposite samples, the moisture absorption and resorption tests were performed in atmospheres with 47%, 73%, and 91% relative humidity at room temperature. Dynamic mechanical analysis was employed to evaluate the hygrothermal ageing effect for unconditioned and environmentally “aged” samples. It was found that moisture sorption was not fully reversible, and the extent of the irreversibility on thermomechanical properties was different for the epoxy and the nanocomposite. The addition of both MWCNTs to the epoxy resin resulted in reduced sorption characteristics for all sorption tests and improved the hydrothermal stability of the cyclic moisture effect on the thermomechanical properties, which was supported by the reduction of the swelling rate after the moisture absorption–desorption cycle and explained by possible reduction of the free volume in the epoxy resin.

However some corrections are needed:

The purity of the reagents taken for synthesis should be given, because it can have a very large impact on it.

The English language should be corrected.

Novelty elements should be better highlighted in the introduction. should be cited in Introduction section; for example:

Hydroxyapatite composites with multiwalled carbon Nanotubes Adsorption Science & Technology 2017, Vol. 35(5–6) 534–544

 Electrokinetic Properties of the Pristine and Oxidized MWCNT Depending on the Electrolyte Type and Concentration   11(1) (2016) Article number 166

Author Response

1. The purity of the reagents taken for synthesis should be given, because it can have a very large impact on it.

No synthesis was performed during the research which was described in the paper. All reagents used for the experiments were of analytical grade purity and were used as received. The solutions used for the humid atmospheres (by using KCNS, NaCl and K2SO4) were prepared with distilled water, and the real value of relative humidity was measured and regularly checked by using Rotronic hygrometer. Moreover, within the same atmosphere the samples were stored in the same dessicator keeping the same conditions during absorption, desorption and resorption. Therefore, the purity of the reagents may have the same and minimal effect on the sorption and thermomechanical properties of the epoxy and nanocomposite discussed in the paper.

2. The English language should be corrected.

English language was edited. The corrections are indicated by using track changes.

3. Novelty elements should be better highlighted in the introduction.

The novelty elements were highlighted in the introduction. Following text was added to the manuscript:

“The novelty of the research was to find the interrelation among the absorption-desorption-resorption characteristics in a wide range of humidity and the thermomechanical properties before and after environmental ageing for the epoxy filled with different MWCNTs”.

Should be cited in Introduction section; for example:

Hydroxyapatite composites with multiwalled carbon Nanotubes Adsorption Science & Technology 2017, Vol. 35(5–6) 534–544.

Electrokinetic Properties of the Pristine and Oxidized MWCNT Depending on the Electrolyte Type and Concentration 11(1) (2016) Article number 166.

One of the suggested references [10] was added to the Introduction regarding the dispersion of the MWCNTs:

“The maximal improvements in barrier characteristics and stability against environmental factors could be obtained by optimizing the dispersion process and the chemical functionalization of MWCNTs in order to reduce the nanoparticles agglomeration, improving at the same time the interfacial strength with surrounding polymer matrix [9]. Moreover, nanotube degree of dispersion depends also on the uniformity of the boundary layer and considered nanotube morphology [10]. For all these reasons, the number of factors affecting the final performance of the NC can be rather considerable leading to contradictory results towards the improvements in barrier and mechanical properties discussed before.”

Reviewer 3 Report

The manuscript entitled "Cyclic Moisture Sorption and its Effects on the Thermomechanical Properties of Epoxy and Epoxy/MWCNT Nanocomposite" by Tatjana Glaskova-Kuzmina and co-workers deals with the characterization of different polymers and polymers/NT composites. The authors focalized their attention to the effect moisture (more precisely adsorption/desorption effect). As explained by the authors, the effect of moisture exposure on the mechanical properties f both polymer and composites is not univocally analysed. 

Yet, in my opinion, the reviewed manuscript lacks for sufficient investigation and scientific soundness. For example authors use two different type of MWCNT with diameter up to 9 or 140 nm; this not allow to evidence a clear effect of the dimension of the NTs. Similar issues could be raised concerning the amount of NTs employed to dope the polymeric matrix or the RH values selected or the specimen dimension.

Additionally, the authors set the experimental temperature at 20°C (RT); indeed, it could be very meaningful to analyse the moisture uptake/desorption over a wider range of temperature (i.e. lower and higher) miming the real temperature condition experimented in operando.

In figure 2, the trend of un-modified epoxy-based material seems to substantially deviate (especially in the plateaux range) from the 3D-Fick's law in A/D/R curves.

The role of "bound" and "unbound" water should be deeply and more quantitatively analysed. The reported description is too qualitatively and it lacks for sufficient experimental proves.

Values reported in table 1 did not present any clear trend of w (%) and εsw (%) with respect to the different amount of NTs added in the polymeric matrix.

Some minor points:

is the preconditioning time really as long as two month? or is it a refuse?

Considering the low scientific soundness and significant and appeal of content, my suggestion is to REJECT the present manuscript

Round 2

Reviewer 3 Report

I appreciated the efforts made by the authors to improve the overall merit of the manuscript according to reviewers' comment. Some point have been solved, some others have not. Please find hereafter additional comments.

POINT 1. I understand that the point of the manuscript is to investigate the moisture A-D-R phenomena with different nanofillers. Yet, the nature of this nanofillers, as well as their dimension and loading, are meaningful parameters that could heavily influence the behaviour of epoxy NCs. Therefore, the authors have to add a paragraph explain their choice of MWCNTs, at least.

POINT 4. As stated by the authors in the reply document, “This deviation is typically observed in the middle part of the curve, when Fick’s model declines with increasing relative humidity of environment, since the diffusion mechanism becomes less dominating, and other mechanisms, such as the interaction between the polymer and the diffusate and/or relaxation processes, start to affect the moisture transport in the polymer”, a clear deviation could be evidenced in the middle range of A,D,R curves. In my opinion, the authors should revise their model (at least for 91% sample) to take into account the effect of raising RH. This is a crucial point of the present manuscript. I agree with the authors that the introduction of non-Fickain models will increase the number of independent variables; yet, I still strongly suggest doing that in order to deeply understand the behaviour of epoxy NCs at high RH conditions.

POINT 5. Even if a hypothetical discussion of bound and free water molecules is in line with the literature, a more quantitative analyses of this phenomenon could be a strong point to enhance the overall method of the manuscript.

POINT 6. Despite of what stated by the authors and the employment of arrows to evidence the trends, I still doubt on the presence of a clear trend in the w (%) and εsw (%) variation with respect to the different amounts of NTs added in the polymeric matrix.

As the authors could se from the figure, I highlighted the trend (with respect to the different amounts of NTs added in the polymeric matrix) with orange arrows. They seems to be quite random (except for 91% samples, whose strongly deviate from Fick’s law)

A trend could be evidenced by considering the increase of RH with fixed concentration and nature of fillers.

In my opinion, the overall merit of the manuscript is higher compared to the first revision round. Yet, it still lacks for complete discussion of the obtained results. My suggestion (after reading the comments of other reviewers too) is to reconsider the manuscript after MAJOR REVISIONS.
